# Driving Through Uncertainty: Risk-Averse Control with LLM Commonsense for Autonomous Driving under Perception Deficits

## Abstract

Partial perception deficits can compromise autonomous vehicle safety by disrupting environmental understanding. Existing protocols typically default to entirely risk-avoidant actions such as immediate stops, which are detrimental to navigation goals and lack flexibility for rare driving scenarios. Yet, in cases of minor risk, halting the vehicle may be unnecessary, and more adaptive responses are preferable. In this paper, we propose LLM-RCO, a risk-averse framework leveraging large language models (LLMs) to integrate human-like driving commonsense into autonomous systems facing perception deficits. LLM-RCO features four key modules interacting with the dynamic driving environment: hazard inference, short-term motion planner, action condition verifier, and safety constraint generator, enabling proactive and context-aware actions in such challenging conditions. To enhance the driving decision-making of LLMs, we construct DriveLM-Deficit, a dataset of 53,895 video clips featuring deficits of safety-critical objects, annotated for LLM fine-tuning in hazard detection and motion planning. Extensive experiments in adverse driving conditions with the CARLA simulator demonstrate that LLM-RCO promotes proactive maneuvers over purely risk-averse actions in perception deficit scenarios, underscoring its value for boosting autonomous driving resilience against perception loss challenges.

## 1 Introduction

Modern autonomous driving systems rely on sensor data inputs from cameras and LiDARs, processed by deep neural networks, to enable perception, understanding, and interaction with the environment (Hu et al., 2023; Liu et al., 2023; Casas et al., 2020; Li et al., 2022). Since perception directly influences driving decisions, partial perception deficits caused by sensor failures or attacks can be fatal, leading to catastrophic consequences (Ceccarelli & Secci, 2022; Min et al., 2023; Shafaei et al., 2018; Wang et al., 2020).

Considering driving scenarios with perception deficits, as illustrated in Figure 1, the loss of safety-critical object information can undermine the driving safety of autonomous agents. Conventional fail-safe protocols default to entirely risk-avoidant actions such as immediate stops. However, not all perception loss are catastrophic to autonomous driving safety, so triggering a fail-safe maneuver for every detected perception loss event is impractical and detrimental to the navigation goals (Antonante et al., 2023; Chakraborty et al., 2024). Moreover, risk-avoidant fail-safe strategies lack the flexibility to handle diverse driving scenarios and rely on fully restored perception inputs before resuming motion, further limiting their effectiveness (Vom Dorff et al., 2020).

Human drivers demonstrate that safe navigation is possible with limited visibility by leveraging commonsense reasoning and driving experience to infer critical visual information (Fu et al., 2024). For example, obstructions in peripheral vision may not impact immediate driving decisions, and cautious proceeding remains viable when maintaining sufficient physical distance from deficit regions or when these deficit regions show temporal stability. This human-inspired insight suggests that autonomous systems should conduct risk-averse, proactive, and context-aware strategies rather than defaulting to overly conservative actions. While commonsense plays a crucial role in human judg-

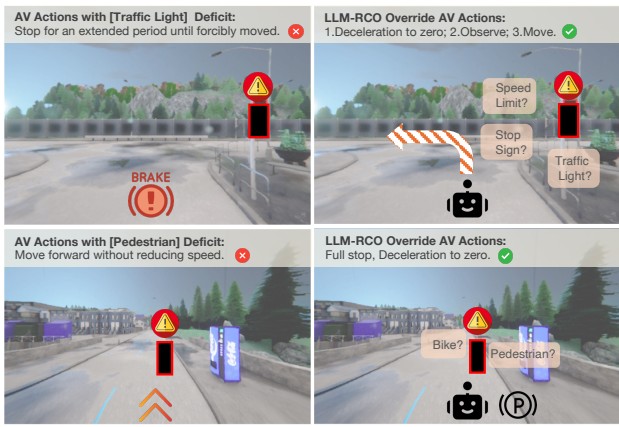

Figure 1: With perception deficits, deep learning-based AV agents take either over-conservative or unsafe actions, while LLM-RCO draws on LLM commonsense to infer possible hazards and plan safe motions.

ment to ensure a smooth and safe driving experience, current autonomous driving research has yet to explore how to integrate commonsense into decision-making under such challenging conditions.

Multimodal large language models (LLMs) have demonstrated remarkable reasoning capabilities and extensive commonsense knowledge across diverse applications (Huang & Chang, 2022; Zhao et al., 2024; Sap et al., 2020). Trained on vast web-scale corpora, multimodal LLMs inherently encode a wealth of traffic rules and driving knowledge, potentially equipping them with a strong driving commonsense. However, effectively leveraging this knowledge for autonomous driving remains an open question. Unlike end-to-end LLM driving agents (Shao et al., 2024; Mao et al., 2023a), which can be trained through imitation learning from rule-based pilots, there is no well-established expert to provide optimal strategies for proactive movement under incomplete perception. Most failsafe mechanisms simply default to simple stops, making it challenging to collect training data for a deep learning model that can guide autonomous vehicle movement with perception deficits.

To address these challenges, we propose LLM-Guided Resilient Control Override (LLM-RCO), a risk-averse framework that harnesses the commonsense and reasoning capabilities of multimodal LLMs to enhance the resilience and safety of autonomous vehicles under partial perception deficits. LLM-RCO overrides autonomous vehicle control to mitigate the risk of hazardous actions resulting from erroneous predictions on compromised perception data, ensuring safer operation in adverse conditions. Drawing from human drivers' reasoning in perception deficit scenarios, we break down this process into four modules in LLM-RCO: (1) *Hazard Inference Module*, which evaluates potential risks in deficit areas using past camera frames; (2) *Short-Term Motion Planner*, which generates a flexible sequence of action-condition pairs tailored to the driving context and hazard inference outcomes; (3) *Action Condition Verifier*, which checks the consistency of information deficits and immediate hazards in current observations to ensure action feasibility for safe and reliable vehicle control; (4) *Safety Constraints Generator*, which sets vehicle control limits based on safety-critical factors like weather, lighting, and traffic density. Additionally, we construct the DriveLM-Deficit dataset based on DriveLM-GVQA (Sima et al., 2023) to fine-tune the LLM for hazard inference and planning, enhancing the subsequent short-term motion planning process.

**Contributions.**

- We propose LLM-RCO, a risk-averse framework leveraging LLM commonsense and reasoning capabilities to enable proactive movement under perception deficits.
- We build DriveLM-Deficit, a dataset having 53, 895 driving videos with perception deficits of safety-critical objects. We benchmark the performance of advanced LLMs on DriveLM-Deficit, highlighting their limitations. We fine-tune Qwen2-VL-2B-Instruct on DriveLM-Deficit using LoRA to enhance hazard inference and motion planning of LLM-RCO.
- We validate LLM-RCO in closed-loop environments using the CARLA (Dosovitskiy et al., 2017) simulator separately with TransFuser (Prakash et al., 2021) and InterFuser (Shao et al., 2022) agents, showing that LLM-RCO consistently enhances driving scores by adopting a cautious yet proactive driving style.

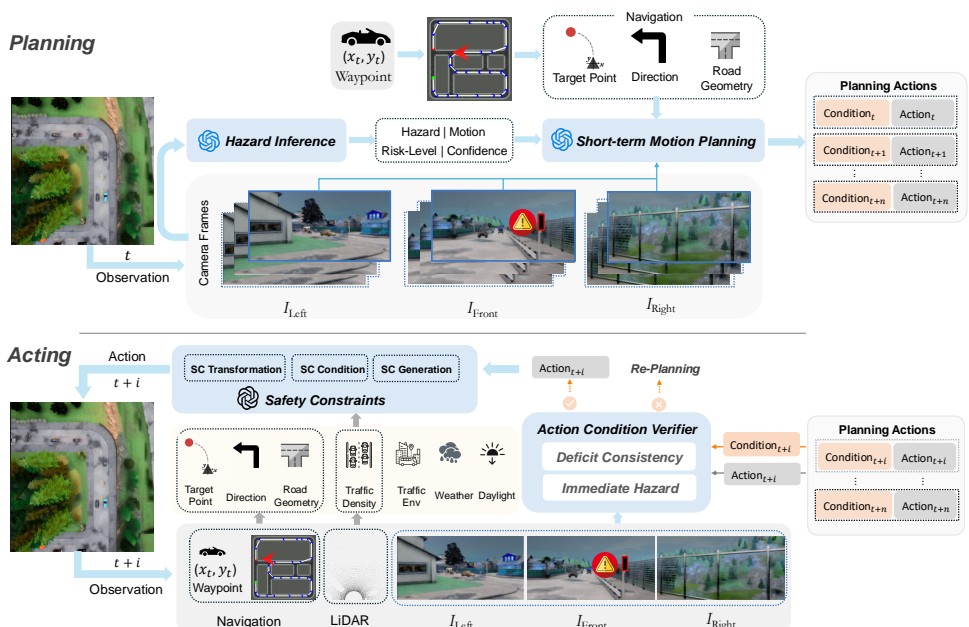

Figure 2: LLM-RCO overview.

## 2 RELATED WORK

**LLM in Autonomous Driving.** Recent work has integrated multimodal LLMs into autonomous driving, improving agent performance in closed-loop settings (Mao et al., 2023a;b; Huang et al., 2024; Long et al., 2024; Shao et al., 2024), enhancing security against adversarial attacks (Kong et al., 2024; Song et al., 2024; Chung et al., 2024; Aldeen et al., 2024b;a), and supporting explainability in decision-making (Chen et al., 2024; Peng et al., 2024; Xu et al., 2024). LLMs also enable commonsense, knowledge-driven decisions when integrated into existing stacks (Cui et al., 2024; Sha et al., 2023; Wang et al., 2024b; Fang et al., 2024). However, motion planning under partial perception deficits, such as attacks or sensor failures, remains an underexplored area where LLMs could further enhance driving resilience.

**Fail-Safe Strategies in Autonomous Driving.** Fail-safe strategies in autonomous driving are typically composite with multi-sensor fusion (Liu et al., 2023; Cao et al., 2021), degraded modes (Magdici & Althoff, 2016; Jiang et al., 2024), and risk-avoidant emergency maneuvers such as stops or pull-overs (Bogdoll et al., 2022; Gao et al., 2021). However, these methods have key limitations: rule-based logic struggles with long-tail events, emergency stops disrupt traffic, and perception-dependent planning remains vulnerable to failures and attacks (Magdici & Althoff, 2016; Ramanagopal et al., 2018; Koopman & Wagner, 2016; Pek & Althoff, 2020). Hard-coded safety rules, such as overriding stops for uncertain traffic light detections, lack adaptability in dynamic conditions (Ren et al., 2019; Yurtsever et al., 2020; Koopman & Wagner, 2017). This motivates LLM-RCO, which aims to provide a more risk-averse and flexible solution for handling perception failures in autonomous driving systems.

## 3 RISK-AVERSE CONTROL: LLM-RCO

In this section, we propose LLM-RCO, which features the incorporation of LLM commonsense in multiple steps of planning and reasoning processes to enhance the resilience and safety of autonomous vehicle control strategies.

**Preliminary.** We define the action of autonomous driving at time $t$ as $\mathcal{A}_t$. Action contains three vehicle control parameters: throttle, brake, and steering wheel, i.e., $\mathcal{A}_t = [throttle_t, brake_t, steer_t] \in [0, 1] \times [0, 1] \times [-1, 1]$. An action sequence is a collection of pairs, each consisting of a tentative candidate action and its execution condition, spanning from time $t$ to time $t + n$, de-

noted as $S_t = \{(\mathcal{C}_t, \mathcal{A}_t), \cdots, (\mathcal{C}_{t+n}, \mathcal{A}_{t+n})\}$. The environmental information at $t$ is $E_t = [Perception_t, Navi_t, Surrounding_t]$, where $Perception_t = \{I_t^L, I_t^F, I_t^R\}$ is the real-time images separately collected by left, front, and right camera sensors; $Navi_t$ is the navigation contains the coordinate of the target point, current direction, and road geometry; and $Surrounding_t$ is the surrounding environmental information that describe traffic, weather, and daylight information. The safety conditions of the vehicle at time $t$ is $SC_t$, describing constraints such as the maximum following distance.

**Framework Overview.** Unlike the LLM-based autonomous driving agents that employ continuous, real-time action inference, our LLM-RCO framework adopts a proactive, plan-ahead strategy. By reducing the frequency of interactions with the LLM, it lowers inference latency, leading to more efficient and smoother driving. Figure 2 shows the LLM-RCO framework, which consists of planning and acting phases.

(1) When planning, if $S_t$ is empty, LLM-RCO updates $n$-steps in $S_t$ based on the perception context and navigation:

$$S_t = F_{\text{plan}}(Navi_t, Percetion_{\{t-k,\cdots,t\}}). \tag{1}$$

LLM-RCO planning first launches a Hazard Inference Module, an LLM agent that takes multiple frames of sensor context information $Perception_{\{t-k,\cdots,t\}}$, and concludes the potential hazard and its movement into $Hazard_t$ in Eq. equation 3. Then, a Short-term Motion Planning Module, another LLM agent takes concluded hazard information $Hazard_t$, navigation information $Navi_t$ and current time perception information $Perception_t$, and then generates a variable-length (we denote as $n$) condition-action sequence as $S_t$.

(2) When acting, LLM-RCO reads condition-action pairs from $S_t$ sequentially, referencing environment information:

$$\begin{cases} (\mathcal{C}_t, \mathcal{A}_t) = F_{\text{act}}(S_t, E_t) \\ S_{t+1} = S_t \setminus \{(\mathcal{C}_t, \mathcal{A}_t)\}. \end{cases} \tag{2}$$

The condition-action pairs will first be verified by the Action Condition Verifier, which determines whether the current environment perception $E_t$ is suitable for executing the condition-action $(\mathcal{C}_t, \mathcal{A}_t)$. If suitable, LLM-RCO executes the action $\mathcal{A}_t$ by the acting phase. If the Action Condition Verifier denies action $\mathcal{A}_t$ or the $S_t$ is empty, then LLM-RCO will start a new round of planning.

## 4 KEY COMPONENTS IN LLM-RCO

**Hazard Inference.** Partial perception loss can severely impact driving when the AV system cannot detect potential hazards in compromised areas. Inferring these hazards in the perception deficit area enables the system to make more informed driving decisions. We define a hazard based on the object and its movement. Moreover, multiple hazards (denoted as $N_t$) may exist simultaneously. Formally:

$$hazard_t = \{[object_i, motion_i]\big|_{i=1}^{N_t}\}. \tag{3}$$

In LLM-RCO, the hazard inference module is essential for ensuring safe driving decisions by compensating for perception loss. To enhance the hazard inference accuracy, we fine-tune LLMs to jointly infer potential hazards within compromised image regions and the corresponding short-term planning strategies. Assuming that sensor failures or attack patterns remain consistent in the short term, tracking the temporal motion of the unaffected surroundings provides vital insights into their evolving dynamics, enabling the LLM to produce more accurate hazard predictions. Consequently, we input a sequence of the past $k$ multi-view images from the cameras, allowing the model to infer potential hazards and plan with historical context.

$$(hazard_t, plan_t) = \text{LLM}_{\text{HI}}(Percetion_{\{t-k,\cdots,t\}}). \tag{4}$$

where $\text{LLM}_{\text{HI}}$ is a multimodal LLM with the specified hazard inference prompt preset (same below).

**Short-term Motion Planing.** Short-term Motion Planner employs two distinct operational strategies $plan$ that emulate human-like adaptability: *Move* and *Stop-Observe-Move*. As demonstrated in Figure 3, under *Move* strategy, the vehicle proceeds cautiously with short-term motions plan ahead,

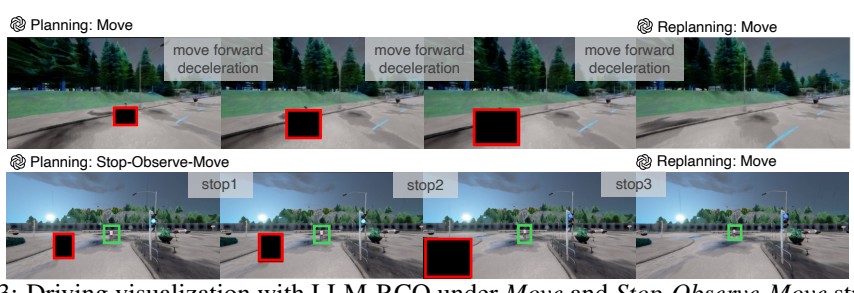

Figure 3: Driving visualization with LLM-RCO under *Move* and *Stop-Observe-Move* strategies.

while *Stop-Observe-Move* requires temporary halting to gather additional information before proceeding to ensure safety in uncertain situations. Under the *Move* strategy, the system generates an adaptive-length sequence of planning steps. For *Stop-Observe-Move* strategy, when the LLM determines that an immediate stop is necessary, it specifies both the waiting duration $wait$ and move trigger conditions $\mathcal{C}_{\text{move}}$ to initiate a new round of motion planning for the vehicle's movement. During the waiting period, only 'stop' actions are added into $S_t$.

$$S_t = \text{LLM}_{\text{SMP}}([hazard_t, plan_t, Navi_t, Perception_t]). \tag{5}$$

Different from conventional fail-safe strategies that only output control actions, LLM-RCO tasks the LLMs to generate an execution condition for each action to enhance driving safety and allow better adaptation to dynamic road conditions. In $S_t$, each element from $(\mathcal{C}_t, \mathcal{A}_t)$ to $(\mathcal{C}_{t+n}, \mathcal{A}_{t+n})$ is a condition-action pair, indicates that the action will be executed only when the condition is met. We define conditions as combinations of 'consistent_deficit' with either 'no_immediate_hazard' or 'immediate_hazard'. When the observed deficits are inconsistent across frames, it implies extreme uncertainty about potential hazards arising from perception deficits. In such cases, the LLM is tasked with real-time motion inference by immediate replanning. Conversely, if deficit patterns remain consistent across frames, the LLM can utilize unaffected perception information and hazard inference results to generate proactive action sequences. As illustrated in Figure 4, under consistent deficits, the LLM may plan more stable actions to enable smooth driving if there is no immediate hazard or opt for more aggressive maneuvers such as sudden brake in the presence of an immediate hazard.

**Action Condition Verifier.** At each timestep, LLM-RCO's acting phase reads and executes the next action $\mathcal{A}_t$ in $S_t$. To ensure the safe execution of plan-ahead actions, we design a rule-based action condition verifier to check $\mathcal{C}_t$ based on real-time perception observation. As the road conditions might be highly dynamic, we introduce a deficit consistency check to enable real-time action inference by motion replanning. As shown in Figure 4, the deficit consistency is evaluated by examining the temporal quantity and spatial shifts of deficits observed across frames. We compare deficit regions across image frames in

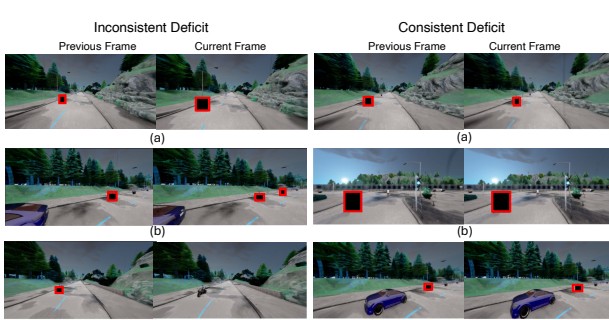

Figure 4: Deficits across frames. **Left:** Scenarios of Inconsistent Deficits: (a) spatial shift, (b) quantity realignment, and (c) deficit disappearance. **Right:** Scenarios of Consistent Deficits: (a) no immediate hazard, (b) immediate hazard due to a large deficit size, (c) immediate hazard from nearby objects in unaffected regions.

terms of both their quantity and relative spatial positions against the background. If the quantity of deficits does not align or if the spatial shift of deficit regions exceeds a predefined threshold, the condition check will fail, prompting instant motion replanning. If the consistency check passes, we assess immediate hazards by analyzing the ratio of deficit regions and detected traffic objects (e.g., cars, trucks, buses, bicycles, pedestrians, motorcycles using YOLOv11) to the camera image. This ratio, computed as the area of deficit regions and traffic object bounding boxes relative to the image size, serves as a hazard proximity indicator. If it exceeds a predefined threshold (set to 0.05), it indicates an immediate hazard near the vehicle.

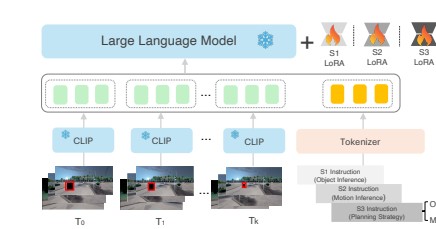 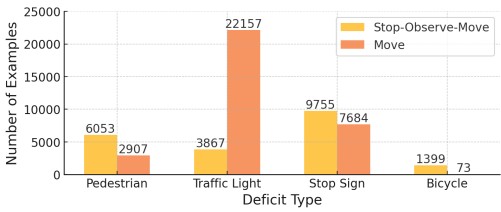

(a) Driving-Oriented Fine-tuning Scheme.  (b) DriveLM-Deficit Dataset.

Figure 5: Fine-tuning on DriveLM-Deficit.

**Safety Constraint Module.** In LLM-RCO, the Safety Constraint Module adjusts the action $\mathcal{A}_t$ from the Action Condition Verifier to ensure compliance with safety constraints. As safety constraints depend on various factors, for example, the maximum following distance in snowy conditions differs from that in normal conditions, we leverage the extensive commonsense knowledge of LLMs to improve driving safety. Specifically, based on the driving context including weather, daylight, traffic density, road geometry, etc., we task LLMs to generate a set of loose safety constraints $\mathcal{SC}_t$ to enhance vehicle control safety. As driving conditions remain stable over short time periods, we don't need real-time inference for $\mathcal{SC}_t$ generation. Due to space limitations, more implementation details are provided in the Appendix A.2.

## 5 DRIVING-ORIENTED FINE-TUNING

To deliver straightforward optimal driving actions to LLM, we propose fine-tuning an LLM for hazard inference and strategy selection to guide subsequent short-term motion planning. The fine-tuning process involves three tasks: (1) Object Inference, which identifies objects within deficit regions using contextual cues from unaffected parts of the image; (2) Motion Inference, which analyzes motion behaviors in deficit regions based on historical perception frames; and (3) Planning Strategy, which decides between "move" or "stop-observe-move" approaches using insights from the previous tasks. For example, the system may choose "move" with the cue "Bicycle, oncoming at a constant speed," or "stop-observe-move" with the cue "Pedestrian, crossing to the other side of the road." As shown in Figure 5a, we use three Low Rank Adaptation (LoRA) (Hu et al., 2022) modules to adapt the model to these tasks.

We process GVQA dataset in DriveLM-Carla (Sima et al., 2023) for fine-tuning, which is collected using PDM-Lite (Beißwenger, 2024) and achieves 100% completion with zero infractions. GVQA dataset comprises 1.6M QA pairs along with sensor data, keyframes, and frame-level QAs covering perception, prediction, and planning. It employs a rule-based annotation pipeline to determine the ego vehicle's braking status using privileged information about objects, hazards, and scene measurements. Building on DriveLM-GVQA, we construct the DriveLM-Deficit dataset tailored to our specific case. DriveLM-Deficit consists of 53, 895 videos, each composed of five frames sampled every two steps, forming a 5-second clip at 1 fps. It includes deficits related to traffic lights, traffic signs, pedestrians, and bicycles for fine-tuning. The statistics of DriveLM-Deficit is shown in Figure 5b, and more details are in Appendix A.3.

## 6 EXPERIMENTS

**Fine-tuning Details.** We choose to supervise fine-tune a lightweight pre-trained vision language model "Qwen2-VL-2B-Instruct" (Wang et al., 2024a) with LoRA using DriveLM-Deficit. we only add LoRA to the "q_proj" and "v_proj" attention layers, and we set the rank to 4 for all experiments.

**LLM-RCO Implementation.** To control the autonomous agent, we define high-level actions with driving behavior (move forward, stop, change lane to left, change lane to right, turn left, turn right) and speed control (constant speed, deceleration, quick deceleration, deceleration to zero, acceleration, quick acceleration), the action tokens are mapped to control parameters in a deterministic manner based on the navigation to the next target point, more details are provided in Appendix A.1.

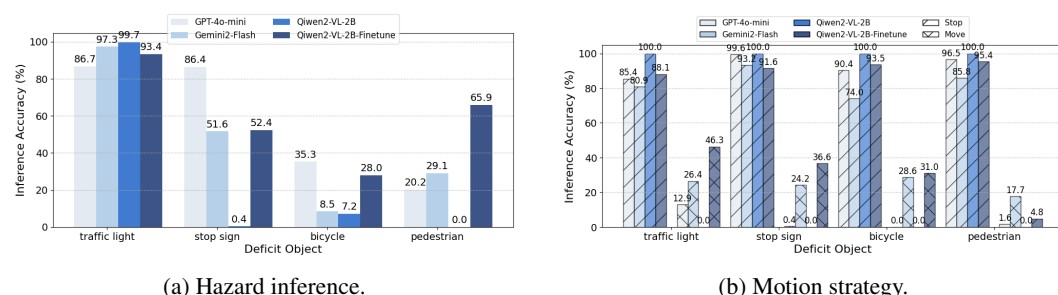

(a) Hazard inference.                      (b) Motion strategy.

Figure 6: Accuracy of various LLM backbones on Drive-Deficit.

We employ GPT-4o-mini (Achiam et al., 2023) and fine-tuned Qwen2-VL-2B as LLM-RCO backbone models.

**Perception Deficit Scenarios.** In the experiments, we simulate different perception deficits of safety-critical objects using YOLO11 (Jocher & Qiu, 2024). We track and mask safety-critical objects in real-time, including traffic lights, stop signs, pedestrians, and bicycles. When these objects are masked out, control transfers to LLM-RCO.

**Carla Simulator.** Our implementation employs CARLA 0.9.10.1 simulator with the Transfuser (TF) and Interfuser (IF) agents, which integrate both image and LiDAR data for end-to-end vehicle control. Transfuser incorporates rule-based stuck detection and employs a force-move command within its control prediction. Similarly, Interfuser features a rule-based force-move mechanism integrated into its safe controller module. In our setup, both force-move mechanisms remain at their original default settings to enable a fair comparison with LLM-RCO. Experiments are conducted on the Longest6 benchmark, which comprises 36 routes averaging 1.5 km in length and featuring high densities of dynamic actors in diverse environmental conditions.

**Metrics.** We adopt three primary metrics employed by the CARLA Leaderboard for evaluations: Route Completion (RC), Infraction Score (IS), and Driving Score (DS). The IS typically accounts for collisions and traffic rule violations, including running red lights and stop signs. In the experiments, we modified the IS calculation correspondingly in the deficit scenarios of traffic lights and traffic signs by excluding red lights and stop sign violations from scoring. The DS is calculated as the product of RC and IS, reflecting both driving progress and safety. We report the metrics for three runs. Additionally, we report the Average Speed (AS) to assess driving efficiency, which is calculated as the total route length divided by the game time. Note that the game time is counted by the ticks in the simulation, and the model inference time is not factored into this count.

## 7 RESULTS

**Fine-tuning results on DriveLM-Deficit.** In Figure 6, we evaluate the hazard inference and motion planning accuracy of GPT-4o-mini, Gemini-2-flash Team et al. (2023), Qwen2-VL, and fine-tuned Qwen2-VL on the Drive-Deficit dataset. As shown in Figure 6a, even advanced LLMs struggle to infer possible objects in occluded or missing image regions under driving scenarios. While they excel at identifying traffic lights and stop signs, they perform poorly in rare cases involving pedestrians and bicycles. To address this limitation, we fine-tuned the open-source Qwen2-VL model for hazard inference and it achieves higher accuracy on both pedestrians and bicycles than GPT-4o-mini and Gemini2-flash. In Figure 6b, we evaluate the LLMs' ability to infer appropriate motion plan based on past camera frames under perception deficits. The results indicate that advanced LLMs often opt to conservative stop even in scenarios where movement is feasible, which is reflected in lower prediction accuracy for "move" motion strategy across all deficit cases.

**Overall Results.** We report the evaluation results of the LLM-RCO against various perception deficits in two experimental settings: without game time limit shown in Table 1 and with 30-min system time limit shown in Table 2. The game time excludes model inference time, and system time accounts for inference time. In Table 1, we report the average speed (AS) to evaluate the stop rate in the AV movement, where a lower AS indicates more frequent stops. The results in both

Table 1: Comparison of driving performance with unlimited game time. † denotes the highest driving score.

| Experiment | Metric | Traffic Light | | Stop Sign | | Pedestrian | | Bicycle | |
|---|---|---|---|---|---|---|---|---|---|
| | | TF | IF | TF | IF | TF | IF | TF | IF |
| NO Visual Deficit | RC | 100 | 100 | 100 | 100 | 100 | 100 | 100 | 100 |
| | IS | 0.13 ± 0.06 | 0.71 ± 0.08 | 0.13 ± 0.06 | 0.71 ± 0.08 | 0.13 ± 0.06 | 0.71 ± 0.08 | 0.13 ± 0.06 | 0.71 ± 0.08 |
| | DS | 13.18 ± 6.17 | 71.30 ± 8.09 | 13.18 ± 6.17 | 71.30 ± 8.09 | 13.18 ± 6.17 | 71.30 ± 8.09 | 13.18 ± 6.17 | 71.30 ± 8.09 |
| | AS | 1.55 ± 0.28 | 2.14 ± 0.07 | 1.55 ± 0.28 | 2.14 ± 0.07 | 1.55 ± 0.28 | 2.14 ± 0.07 | 1.55 ± 0.28 | 2.14 ± 0.07 |
| Visual Deficit LLM-RCO ✗ | RC | 100 | 94.30 ± 5.01 | 100 | 95.39 ± 5.26 | 100 | 94.99 ± 5.01 | 100 | 100 |
| | IS | 0.05 ± 0.04 | 0.29 ± 0.05 | 0.02 ± 0.01 | 0.23 ± 0.07 | 0.003 ± 0.001 | 0.16 ± 0.08 | 0.07 ± 0.02 | 0.36 ± 0.07 |
| | DS | 4.88 ± 3.76 | 27.44 ± 3.43 | 2.18 ± 0.97 | 21.20 ± 5.67 | 0.32 ± 0.11 | 15.12 ± 6.48 | 6.54 ± 2.10 | 36.47 ± 6.93 |
| | AS | 1.76 ± 0.07 | 1.78 ± 0.01 | 1.87 ± 0.02 | 2.44 ± 0.23 | 1.95 ± 0.21 | 1.05 ± 0.13 | 1.88 ± 0.07 | 2.27 ± 0.06 |
| Visual Deficit LLM-RCO ✓ HI-GPT,STP-GPT | RC | 94.53 ± 0.24 | 98.31 ± 4.28 | 100 | 100 | 100 | 100 | 100 | 94.10 ± 3.45 |
| | IS | 0.11 ± 0.05 | 0.32 ± 0.06 | 0.03 ± 0.01 | 0.22 ± 0.07 | 0.019 ± 0.004 | 0.21 ± 0.04 | 0.09 ± 0.03 | 0.43 ± 0.07 |
| | DS | 10.97 ± 3.96 | 30.70±5.53† | 2.67±0.88† | 21.19 ± 6.84 | 1.87 ± 0.37 | 19.99±4.76† | 9.07 ± 2.71 | 41.80±6.95† |
| | AS | 1.97 ± 0.04 | 1.89 ± 0.04 | 1.94 ± 0.02 | 1.72 ± 0.03 | 1.48 ± 0.03 | 0.94 ± 0.17 | 1.40 ± 0.07 | 0.86 ± 0.01 |
| Visual Deficit LLM-RCO ✓ HI-Qwen,STP-GPT | RC | 94.15 ± 0.33 | 99.10 ± 5.15 | 100 | 100 | 100 | 95.30 ± 4.17 | 100 | 95.41 ± 0.11 |
| | IS | 0.12±0.04 | 0.31 ± 0.06 | 0.03±0.01 | 0.23 ± 0.07 | 0.023±0.006 | 0.19 ± 0.03 | 0.09 ± 0.02 | 0.42 ± 0.08 |
| | DS | 11.18±3.87† | 30.4 ± 5.66 | 2.59±0.73 | 22.65±6.91† | 2.34±0.64† | 18.71±4.05 | 9.13±2.45† | 41.54 ± 8.10 |
| | AS | 2.08 ± 0.03 | 1.82 ± 0.06 | 1.96 ± 0.03 | 1.73 ± 0.02 | 1.15 ± 0.02 | 0.92 ± 0.33 | 1.42 ± 0.08 | 0.97 ± 0.03 |

Table 2: Comparison of driving performance with a 30-minute system time limit. † denotes the highest driving score.

| Experiment | Metric | Traffic Light | | Stop Sign | | Pedestrian | | Bicycle | |
|---|---|---|---|---|---|---|---|---|---|
| | | TF | IF | TF | IF | TF | IF | TF | IF |
| NO Visual Deficit | RC | 97.35 ± 3.89 | 99.13 ± 1.41 | 97.35 ± 3.89 | 99.13 ± 1.41 | 97.35 ± 3.89 | 99.13 ± 1.41 | 97.35 ± 3.89 | 99.13 ± 1.41 |
| | IS | 0.14 ± 0.06 | 0.71 ± 0.03 | 0.14 ± 0.06 | 0.71 ± 0.03 | 0.14 ± 0.06 | 0.71 ± 0.03 | 0.14 ± 0.06 | 0.71 ± 0.03 |
| | DS | 13.52 ± 4.93 | 70.62 ± 5.15 | 13.52 ± 4.93 | 70.62 ± 5.15 | 13.52 ± 4.93 | 70.62 ± 5.15 | 13.52 ± 4.93 | 70.62 ± 5.15 |
| Visual Deficit LLM-RCO ✗ | RC | 30.19 ± 5.32 | 44.90 ± 0.55 | 37.03 ± 0.18 | 45.33±0.21 | 37.11±0.06 | 42.20 ± 1.07 | 38.22±0.04 | 44.70 ± 0.93 |
| | IS | 0.6±0.11 | 0.63±0.09 | 0.14 ± 0.03 | 0.68 ± 0.10 | 0.20 ± 0.05 | 0.36 ± 0.09 | 0.17 ± 0.08 | 0.70 ± 0.06 |
| | DS | 17.89 ± 3.53 | 31.43 ± 3.07 | 5.0 ± 1.16 | 32.73 ± 3.89 | 7.48 ± 1.87 | 16.12 ± 3.82 | 6.43 ± 3.11 | 31.29 ± 2.82 |
| Visual Deficit LLM-RCO ✓ HI-GPT,STP-GPT | RC | 58.57±3.07 | 56.11 ± 1.92 | 37.17±0.45 | 43.15 ± 1.22 | 36.20 ± 0.16 | 41.10 ± 1.32 | 32.50 ± 0.42 | 42.92 ± 0.91 |
| | IS | 0.32±0.06 | 0.60 ± 0.07 | 0.24 ± 0.03 | 0.78±0.09 | 0.28±0.02 | 0.42±0.09 | 0.80 ± 0.13 | 0.78±0.07 |
| | DS | 18.58±3.51† | 33.43 ± 4.18 | 8.95 ± 0.72 | 33.70±4.09† | 9.94±0.81 | 18.79±4.66† | 26.26±4.† | 32.88 ± 3.01 |
| Visual Deficit LLM-RCO ✓ HI-Qwen,STP-GPT | RC | 61.02±4.88 | 58.44±2.31 | 36.39 ± 0.87 | 43.60 ± 1.27 | 40.12±0.70 | 44.81±1.33 | 30.52 ± 0.62 | 46.80±1.21 |
| | IS | 0.28 ± 0.04 | 0.58 ± 0.04 | 0.29±0.03 | 0.74 ± 0.08 | 0.28 ± 0.03 | 0.41 ± 0.08 | 0.81±0.07 | 0.76 ± 0.06 |
| | DS | 17.13 ± 2.44 | 33.90±2.34† | 10.60±1.17† | 32.26 ± 3.51 | 11.03±1.66† | 18.27 ± 4.40 | 25.02 ± 2.94 | 35.60±4.56† |

settings reveal that perception deficits significantly impair autonomous driving performance. For Transfuser, the average speed steadily increases with perception deficits, indicating that the loss of critical hazard information leads to inattentive driving. In contrast, for the Interfuser, we observe that the loss of traffic light information results in overly conservative driving due to incorrect traffic light state predictions, causing it to remain stationary until forcibly moved by its rule-based safe controller.

When LLM-RCO intervenes in vehicle control, as shown in Table 1, we observe that the driving score shows improvement with little compromised route completion score with unlimited game time. In stop sign, pedestrian, and motorcycle cases, the ego vehicle makes more stops, reflected by a decrease in average speed. This cautious driving control of LLM-RCO leads to a higher driving score. For traffic light scenarios, LLM-RCO increases the average speed by reducing unnecessary stops, resolving issues where Interfuser's incorrect traffic light predictions cause vehicles to remain stuck until forcibly moved. Moreover, considering the inference time cost of LLM, as shown in Table 2, despite the lower infraction score of LLM-RCO in the traffic light deficit scenario, where the longer completed routes increase exposure to potential risks for transfuser, the infraction and driving scores of LLM-RCO present a consistent increase in all other perception deficit scenarios. These results demonstrate the fine-tuned Qwen with more accurate hazard inference and motion planning can effectively reduce stops, enabling more proactive movements.

**Speed Control Analysis.** In Figure 7, we report the percentage of various speed control decisions of LLM-RCO against various perception deficits. The results are from Transfuser on route0 of Town01 in longest6. The result shows that the percentage of different speed control remains relatively consistent across different deficit scenarios: approximately 50% of actions maintain a constant speed, around 40% involve

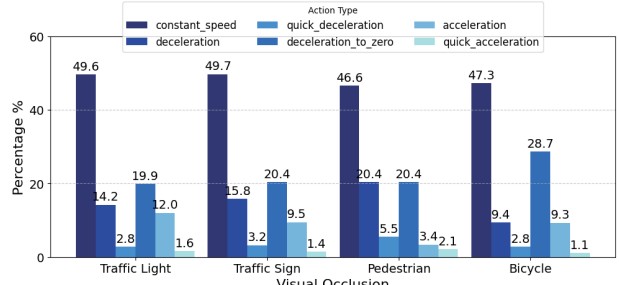

Figure 7: Speed control analysis.

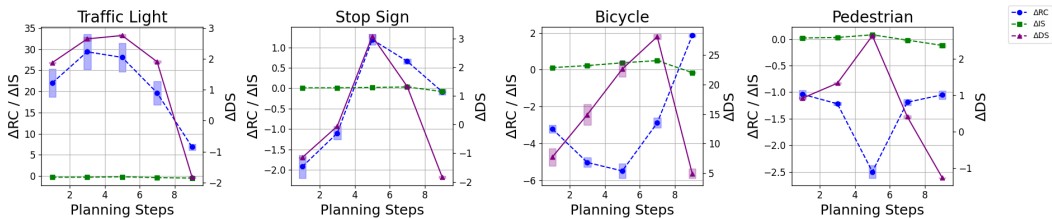

Figure 8: LLM-RCO performance under different action step length limit. Higher $\Delta RC$, $\Delta IS$, and $\Delta DS$ indicate better performance.

deceleration, and only about $10\%$ involve acceleration, indicating that LLM-RCO adopts a cautious driving style, prioritizing safety. Additionally, we observe that the percentage of acceleration actions is lowest in the pedestrian masking scenario, in accordance with rigorous human safety.

**Short-term Motion Planning Analysis.** In Figure 8, we examine the performance of LLM-RCO with different limits on planning step lengths. Within the LLM-RCO framework, a maximum length limit is applied to short-term motion planning, enabling the LLM to reason through a variable number of actions based on the current driving context. We report the evaluation metric change with and without LLM-RCO intervention in Transfuser. Our results reveal that extending planning steps increases accident exposure and lowers infraction scores. While a moderate increase in the step limit can improve driving scores, infraction scores drop sharply at higher limits, indicating that excessive planning steps ultimately degrade performance. We also observe that LLM-RCO exhibits different control strategies for static objects (Traffic Lights, Stop Signs) versus moving objects (Bicycles, Pedestrians). When facing static objects, LLM-RCO tends to execute more proactive maneuvers, reflected in higher average speeds and greater route completion (see Tables 1 and 2). Moderately enlarging the planning step limit can further improve driving scores with more route completion. For moving objects, LLM-RCO typically makes conservative stops, which lowers average speeds and hinders route completion. Slightly extending the step length limit prompts the agent to make additional stops, thereby increasing driving scores with increasing infraction scores.

**Ablation Study.** In Table 3, we conduct ablation studies over the LLM-RCO components based on Transfuser and Interfuser: the Safety Constraint Generator (SC), Hazard Inference Module (HI), Short-Term Motion Planner (ST), and Action Condition Verifier (AC). We report the ablation results in the scenario of motor-

Table 3: Ablation study on the LLM-RCO module design.

| LLM-RCO | | | | TF | | | IF | | |
|---|---|---|---|---|---|---|---|---|---|
| HI | ST | AV | SC | RC ↑ | IS ↑ | DS ↑ | RC ↑ | IS ↑ | DS ↑ |
| ✗ | ✗ | ✗ | ✗ | 37.14 | 0.13 | 5.01 | 44.70 | 0.70 | 31.29 |
| ✗ | ✓ | ✗ | ✗ | **36.35** | 0.25 | 9.16 | 41.27 | 0.70 | 28.89 |
| ✓ | ✓ | ✗ | ✗ | **36.35** | 0.42 | 15.27 | 41.19 | 0.75 | 30.89 |
| ✓ | ✓ | ✓ | ✗ | 34.36 | 0.77 | 26.50 | **43.27** | 0.77 | **33.31** |
| ✓ | ✓ | ✓ | ✓ | 33.77 | **0.80** | **27.08** | 42.92 | **0.78** | 32.88 |

cycle deficit. The results illustrate the contribution of each module to overall driving performance and safety. The test demonstrates that full configuration with all modules active achieves the highest infraction and driving scores. Especially, the core modules of LLM-RCO (HI, ST, AV) are especially necessary to enable valid control override.

# 8 CONCLUSIONS

In this paper, we address a new and challenging problem: enabling proactive safe driving under perception deficits with LLM commonsense and reasoning capabilities. As not all perception deficits are catastrophic to autonomous driving safety, this highlights the need for AVs to be risk-averse, but not entirely risk-avoidant. Therefore, we propose a risk-averse vehicle control framework, LLM-RCO, which incorporates several novel features: action-condition pair generation for verifying action safety with human-crafted rules, a plan-ahead short-term motion planning strategy to reduce latency and inference costs, and safety constraints generation to limit vehicle control based on factors like weather, lighting, and traffic density. Experimental validation demonstrates LLM-RCO's performance to enable proactive movement despite visual data loss, underlining its potential for modern autonomous driving systems.

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

## A  APPENDIX

### A.1  AUTONOMOUS VEHICLE CONTROL

The control parameters in the CARLA simulator include steering, throttle, and brake. We define action as a combination of driving behavior (e.g., move forward, stop, change lane to the left, change lane to the right, turn left, or turn right) and speed control (e.g., constant speed, deceleration, quick deceleration, deceleration to zero, acceleration, or quick acceleration). To translate speed control tokens into CARLA control parameters, we apply the mapping outlined in Table 4. For driving behaviors that influence the steering angle, we use CARLA's PIDController of Autopilot to determine the exact steer parameter based on the ego vehicle's relative position to the next navigation target point. If the LLM-generated action involves a direction change instruction in driving behavior and it aligns with the moving direction on the navigation map, the steering parameters are subsequently computed by the autopilot.

| Speed Control | Carla Control Parameter |
|---|---|
| constant speed | $throttle_t = 0.7, brake_t = 0$ |
| deceleration | $throttle_t = max(0, throttle_{t-1} - 0.2), brake_t = 0.2$ |
| quick deceleration | $throttle_t = max(0, throttle_{t-1} - 0.4), brake_t = 0.4$ |
| deceleration to zero | $throttle_t = 0, brake_t = 0.8$ |
| acceleration | $throttle_t = min(1, throttle_{t-1} + 0.2), brake_t = 0$ |
| quick acceleration | $throttle_t = min(1, throttle_{t-1} + 0.4), brake_t = 0$ |

Table 4: Mapping of speed control commands to vehicle control.

### A.2  SAFETY-CONSTRAINTS GENERATION

The Safety Constraint Module adjusts the action $\mathcal{A}_t$ transported from the Action Condition Verifier to meet the safety constraints in conjunction with measurements of the current vehicle's driving conditions. To accomplish this, the Safety Constraint Module obtains the precise measurements of the vehicle's speed $v$, longitudinal acceleration $a_x$, yaw rate $\omega_z$, and distance from the vehicle in front $d_{follow}$ through built-in Inertial Measurement Unit and Speedometer. If these measurements satisfy the Trigger Condition in Table 5, we transform the action accordingly so that our vehicle meets the appropriate safety constraints. We utilize LLM to gauge the current safety constraints $SC_t = \{v_{max}, d_{min}, ac_{max}, de_{max}, \psi_{max}, d_{brake}\}$ as in Table 5 in conjunction with navigation information $Navi_t$, surrounding information $Surrounding_t$ and LiDAR sensor information $I_t^{LiDAR}$:

$$SC_t = \text{LLM}_{SC}([Navi_t, Surrounding_t, I_t^{LiDAR}]) \qquad (6)$$

For each safety constraint, when the trigger condition is satisfied, we perform the corresponding transformation on the corresponding action $\mathcal{A}_t$'s manifold ($throttle_t$, $brake_t$ or $steer_t$). We set two hyperparameters $\Delta_{throttle}$ and $\Delta_{brake}$ here, as the transformation factors for the throttle and the brake specifically. Thus, we get the final action manifold:

$$\hat{throttle}_t = throttle_t - \mathbf{1}_{v \geq v_{max}} \cdot \Delta_{throttle}$$
$$- \mathbf{1}_{d_{follow} < d_{min}} \cdot \Delta_{throttle} \qquad (7)$$
$$- \mathbf{1}_{a_x > ac_{max}} \cdot \Delta_{throttle} \cdot (a_x - ac_{max}),$$

$$\hat{brake}_t = brake_t + \mathbf{1}_{\frac{v^2}{2 \times de_{max}} > d_{brake}} \cdot \Delta_{brake}$$
$$+ \mathbf{1}_{a_x < -de_{max}} \cdot \Delta_{brake} \cdot (a_x - de_{max}), \qquad (8)$$

$$\hat{steer}_t = steer_t \times [1 - \mathbf{1}_{|\omega_z| > \psi_{max}} \cdot (1 - \frac{\psi_{max}}{|\omega_z|})] \qquad (9)$$

Compositing Eq. equation 7, equation 8, and equation 9, we arrive at the final action $\mathcal{A}_t$ that is executed at time $t$:

$$\mathcal{A}_t = [\hat{throttle}_t, \hat{brake}_t, \hat{steer}_t] \qquad (10)$$

| Safety Constraint | Trigger Condition | Action Transformation |
|---|---|---|
| Max Speed Limit | $v \geq v_{\max}$ | $throttle_t - \Delta_{\text{throttle}}$ |
| Min Following Distance | $d_{\text{follow}} < d_{\min}$ | $throttle_t - \Delta_{\text{throttle}}$ |
| Max Acceleration Limit | $a_{\text{x}} > ac_{\max}$ | $throttle_t - \Delta_{\text{throttle}} \cdot (a_{\text{x}} - ac_{\max})$ |
| Max Deceleration Limit | $a_{\text{x}} < -de_{\max}$ | $brake_t - \Delta_{\text{brake}} \cdot (-de_{\max} - a_{\text{x}})$ |
| Max Yaw Rate Limit | $|\omega_z| > \psi_{\max}$ | $steer_t \times \frac{\psi_{\max}}{|\omega_z|}$ |
| Min Braking Distance | $\frac{v^2}{2 \times de_{\max}} > d_{\text{brake}}$ | $brake_t + \Delta_{\text{brake}}$ |

Table 5: Safety constraints and corresponding transformation of autonomous vehicle control parameters.

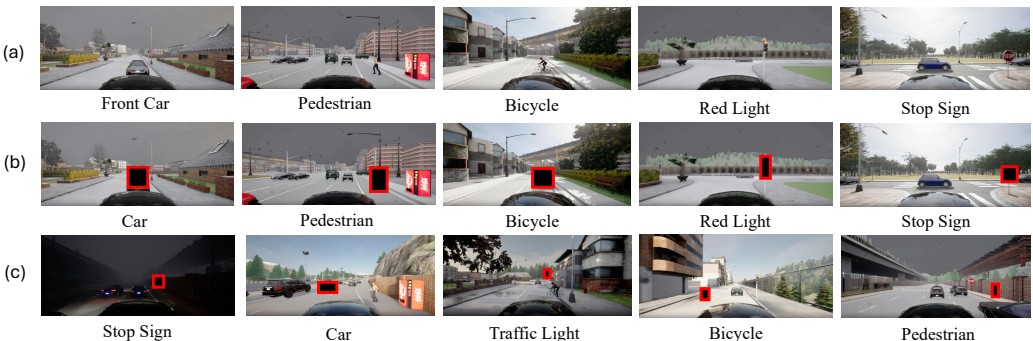

Figure 9: Examples of brake in DriveLM-GVQA and 'Stop-Observe-Move' in DriveLM-Deficit.

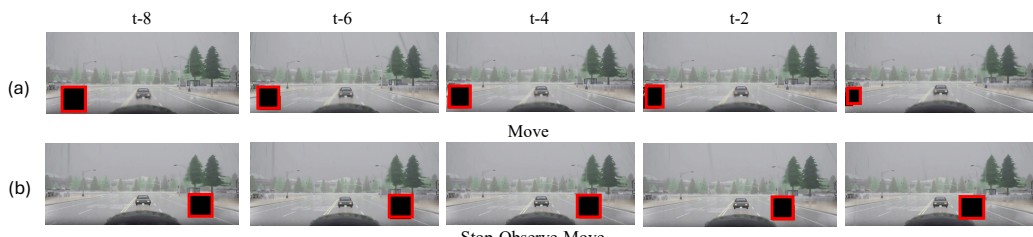

Figure 10: Examples of 'Move' and 'Stop-Observe-Move' motion strategies when the front bicycle is not visible in DriveLM-Deficit.

### A.3 DRIVELM-DEFICIT DATASET

To fine-tune an LLM for hazard inference and motion planning under perception deficits, we post-process DriveLM-GVQA to create DriveLM-Deficit. Specifically, we construct videos using 5 consecutive camera frames from DriveLM-GVQA. The hazard detection and braking annotations of DriveLM-GVQA at the final frame are used to derive the hazard motion and planning strategy label, either "move" or "stop-observe-move". If the last frame indicates braking, we assign the "stop-observe-move" label; otherwise, we set it to "move". As shown in Figure 9 (a), in DriveLM-GVQA, the detected hazards are used to determine the brake status for question "Does the ego vehicle need to brake? Why?" For example, "the ego vehicle should stop because of the traffic light that is red." or "The ego vehicle should stop because of the pedestrian that is crossing the road." Otherwise, the answer is: "There is no reason for the ego vehicle to brake." which corresponds to "move" strategy in our case.

To construct DriveLM-Deficit, we utilize YOLOv11 to track and occlude critical objects in the video, considering two occlusion scenarios. First, as shown in Figure 9 (b), we directly occlude labeled hazards from DriveLM-GVQA, such as a "stop sign." In this case, the rationale for the "stop-observe-move" label is: "The ego vehicle should stop because the invisible region may contain a stop sign." Second, as shown in Figure 9 (c), we occlude other objects, such as a "pedestrian" near a stop sign. In this scenario, the ego vehicle must still stop due to the stop sign, regardless of any hazards inferred in the occluded region. We show two data examples of "move" or "stop-observe-move" planning strategy with occluded bicycle in the perception in Figure 10.

## A.4 Prompt and Output Examples

**Prompting LLM for Safety Constraints Generation**

**System Prompt**

You are a driving assistant to ensure safe driving in Carla simulator. The following context describes the current driving conditions:
1. Weather: #weather
2. Daylight: #daylight
2. Traffic Environment: #traffic_env
3. Road Geometry: #road_geometry
4. Navigation Direction: #direction
5. Relative Position: The relative position of the next moving target point to the location of ego vehicle is #target_point.
6. Traffic Density: There are #traffic_density moving objects within 50 meters of the ego vehicle.
Based on the above context, please provide the following vehicle control constraints to ensure safe driving:
1. Max Speed Limit (km/h)
2. Minimum Following Distance (seconds)
3. Max Acceleration Rate Limit (m/s²)
4. Max Deceleration Rate Limit (m/s²)
5. Max Yaw Rate Limit (degrees/second)
6. Minimum Safe Braking Distance (m)

**Example Values**

#weather>> MidRain    #daylight>>Sunset    #traffic_env >>Urban   #direction >> straight
#target point >> (x, y) = (-0.1240, -58.6818)  #road geometry >> straight 2-way road  # traffic_density >> 12

**LLM Answer**

"Max Speed Limit (km/h)": 40,   "Minimum Following Distance (seconds)": 3,   "Max Acceleration Rate Limit (m/s²)": 1.5,   "Max Deceleration Rate Limit (m/s²)": 3.0,   "Max Yaw Rate Limit (degrees/second)": 10,   "Minimum Safe Braking Distance (m)": 25

**Prompting LLM for Hazard Inference**

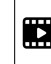 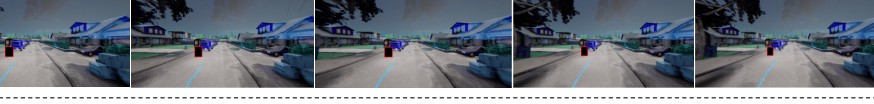

**System Prompt**

You are a Vision Language Model specializing in interpreting visual data from driving images.
Your task is to analyze the provided driving image frames and respond to queries with concise answers follow the answer format defined in the queries. Focus on delivering accurate, succinct answers based on the visual information. Avoid additional explanation unless necessary.

**Task Query Prefix**

This is a 5s video recorded by a front-facing camera mounted on a moving ego vehicle in CARLA. Certain regions of the frames are occluded with black masks and outlined with red bounding boxes to indicate areas with perception deficits.

**Task Specification**

1. Hazard Object Inference
The black mask areas with red bounding boxes in the image are invisible. You need to infer the possible object in the invisible region. Return only one object, for example "traffic light".

2. Hazard Motion Inference
The black mask areas with red bounding boxes in the image are invisible. You need to infer the motion behavior of the invisible region. For example, "it is is crossing paths with the ego vehicle" or "It is approaching from the opposite lane".

3. Planning Strategy Inference
The black mask areas with red bounding boxes in the image maybe a {Object}, which is {Motion}. You need to infer whether the ego vehicle should stop or move. Return your answer with either "move" or "stop" and a brief rationale explaining your answer.

**LLM Answer**

[Motorcycle]    [Oncoming from the opposite lane]    [Move, The motorcycle is oncoming from the opposite lane, meaning it is not directly in the ego vehicle's path. As long as the ego vehicle remains in its lane and there are no other obstacles, it can continue moving safely.]

Figure 11: Hazard inference example.

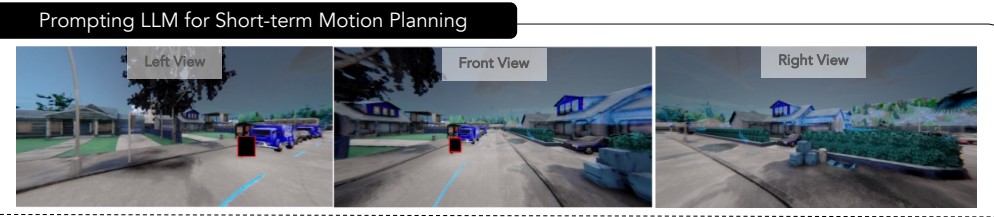

**Prompting LLM for Short-term Motion Planning**

**System Prompt**

You are a Vision Language Model specializing in interpreting visual data from driving images.
Your task is to analyze the provided driving image frames and respond to queries with concise answers follow the answer format defined in the queries. Focus on delivering accurate, succinct answers based on the visual information. Avoid additional explanation unless necessary.

**Task Query for Move**

I have uploaded the left, front, and right driving views captured by the cameras on the ego car in CARLA. The black masks with red bounding boxes indicates the perception deficits. Given the following context:
1. The deficit region maybe a #hazard_object, which is #hazard_motion.
2. Follow the route planner, your current navigation direction is #direction. The relative position between the next navigation waypoint and your current position is #target point. The current road geometry is #road geometry.
3. The speed of ego vehicle is #ego_speed.
Your task is to generate several steps of motion planning for the ego car to continue driving. The maximum step length is #planning length limit. Please analyze the potential hazards near the ego car and the distance to the deficit region to decide the length of the action steps that can be planned ahead. Please give the conditions of left, front, and right driving views to execute the next action.
You need to include the conditions of all views for action generation:
Condition option: ['no_immediate_hazard', 'immediate_hazard'].
The action should be the best combination of driving behavior and speed estimation from the following options:
Driving behavior: ['move_forward', 'change_lane_to_left','change_lane_to_right','turn_left','turn_right','stop'],
Speed estimation: ['constant_speed', 'deceleration', 'quick_deceleration', 'deceleration_to_zero', 'acceleration', 'quick_acceleration'].
Please adapt the steer to follow the navigation. Be careful with the lane-changing.

**Task Query for Stop-Observe-Move**

I have uploaded the left, front, and right driving views captured by the cameras on the ego car in CARLA. The black masks with red bounding boxes indicates the perception deficits. The deficit region maybe a #hazard_object, which is #hazard_motion. You need to complete the motion planning for the ego car by stopping first, and then cautiously continuing to drive. Please give the the least stop steps (maximum is #wait_limit).

**Example Values**

#hazard_motion>>motorcycle    #hazard_object>> oncoming from the opposite lane   #ego_speed >>10.8m/s   #direction >> straight    #target point >> (x, y) = (-0.1240, -58.6818)   #road geometry >> straight 2-way road   #planning length limit >> 5

**LLM Answer**

[('no_immediate_hazard', 'no_immediate_hazard', 'no_immediate_hazard'), ('move_forward', 'constant_speed')],
[('no_immediate_hazard', 'no_immediate_hazard', 'no_immediate_hazard'), ('move_forward', 'constant_speed ')],
[('no_immediate_hazard', 'no_immediate_hazard', 'no_immediate_hazard'), ('move_forward', 'deceleration')],
[('immediate_hazard', 'no_immediate_hazard', 'no_immediate_hazard'), ('stop', 'quick_deceleration')]
[('immediate_hazard', 'immediate_hazard', 'no_immediate_hazard'), ('stop', 'deceleration_to_zero')]

Figure 12: Short-term motion planning example.

A.5  LLM USAGE

In compliance with ICLR 2026 guidelines, we disclose the role of LLMs in preparing this paper. LLMs are involved in our methodology design, dataset construction, and experimental implementation of this paper.

