# OpenReview forum: "Driving Through Uncertainty: Risk-Averse Control with LLM Commonsense for Autonomous Driving under Perception Deficits"
_ICLR.cc/2026/Conference — ICLR 2026 Conference Withdrawn Submission_

### Official Review · Reviewer_LKNa · 2025-10-27

**Soundness:** 2
**Presentation:** 3
**Contribution:** 2
**Rating:** 4
**Confidence:** 3

**Summary:**

The paper proposes LLM-RCO, a large-language-model-guided control override system for autonomous vehicles operating with partial perception failures. Instead of relying on conventional “fail-safe” policies that fully stop the vehicle, LLM-RCO leverages multimodal LLM reasoning to infer possible hazards and generate cautious, short-term motion plans. The framework consists of four modules—hazard inference, short-term motion planning, action-condition verification, and safety-constraint generation—and is fine-tuned on a new dataset (DriveLM-Deficit). Experiments in CARLA show that LLM-RCO can produce more proactive driving behaviors than purely risk-avoidant baselines when parts of the environment are occluded or missing.

**Strengths:**

The problem is clearly important: handling partial perception failure is a major open challenge for real-world autonomous driving systems. The idea of using LLMs for commonsense reasoning to mimic human-like fallback behavior under uncertainty is novel at a conceptual level and well-motivated. The authors also make a useful contribution by releasing a domain-specific dataset (DriveLM-Deficit) and performing systematic simulation studies across multiple types of perception loss.

**Weaknesses:**

- [Lack of formal safety guarantee] Although the paper consistently emphasizes “safety” and “risk-averse control,” the proposed system provides no formal safety assurance—only heuristic checks (e.g., YOLO-based proximity ratios and rule-based thresholds). In safety-critical domains such as autonomous driving, existing frameworks can guarantee constraint satisfaction and safe motion through verifiable methods such as control-barrier-function (CBF) safety filters [1], Hamilton-Jacobi based runtime-assurance/Simplex architectures [2], and reachability-based tracking and shielding [3]. These approaches maintain provable safety envelopes while allowing the vehicle to continue moving, whereas LLM-RCO offers no theoretical bound on risk or constraint violation. Thus, despite its empirical promise, the method cannot be considered safe in the formal sense.


- [Limited novelty and incomplete related-work coverage] Technique-wise, the approach is mainly a domain-specific fine-tuning of a vision-language model combined with rule-based gating, which does not introduce new learning formulations or theoretical insights. Moreover, the paper ignores major prior lines of research on verified policy switching and degraded-mode control, such as the Simplex and runtime-assurance frameworks [2][4] and fail-safe motion planning with online verification [1][3]. These works already address the same goal—maintaining safe, limited operation under perception or planner uncertainty—using provable mechanisms rather than heuristic LLM reasoning. The omission weakens the novelty claim and gives an incomplete view of the state of the art.


[1] Magdici, Silvia, and Matthias Althoff. "Fail-safe motion planning of autonomous vehicles." 2016 IEEE 19th International Conference on Intelligent Transportation Systems (ITSC). IEEE, 2016.

[2] Wabersich, Kim P., et al. "Data-driven safety filters: Hamilton-jacobi reachability, control barrier functions, and predictive methods for uncertain systems." IEEE Control Systems Magazine 43.5 (2023): 137-177.

[3] Herbert, Sylvia L., et al. "FaSTrack: A modular framework for fast and guaranteed safe motion planning." 2017 IEEE 56th Annual Conference on Decision and Control (CDC). IEEE, 2017.

[4] Mehmood, Usama, et al. "The black-box simplex architecture for runtime assurance of autonomous CPS." NASA formal methods symposium. Cham: Springer International Publishing, 2022.

**Questions:**

Please discuss the weaknesses above, i.e., how the safety guarantees and how to position the paper in relation to a long line of Simplex/runtime-assurance frameworks.

---

### Official Review · Reviewer_dJ99 · 2025-10-31

**Soundness:** 3
**Presentation:** 3
**Contribution:** 2
**Rating:** 4
**Confidence:** 4

**Summary:**

This paper proposes LLM-RCO (LLM-Guided Resilient Control Override), a risk-averse control framework for autonomous driving under partial perception deficits. The central idea is to use the commonsense reasoning capabilities of large language models (LLMs) to guide safe, proactive driving decisions when perception sensors (e.g., cameras) fail to detect critical objects.

The architecture includes four modules:

> Hazard Inference, predicting potential unseen hazards from past frames;

> Short-Term Motion Planner, generating conditional action sequences (“move” vs “stop-observe-move”);

> Action Condition Verifier, ensuring consistency and real-time safety;

> Safety Constraint Generator, setting adaptive control bounds based on traffic, weather, and daylight.

The authors introduce DriveLM-Deficit, a dataset of 53,895 clips simulating perception deficits of safety-critical objects, derived from DriveLM-GVQA. They fine-tune Qwen2-VL-2B-Instruct using LoRA on object, motion, and planning tasks, and evaluate the system using CARLA with TransFuser and InterFuser agents. Results show improved driving scores and reduced infraction rates across deficit scenarios, with detailed ablation and sensitivity analyses.

**Strengths:**

+ Addresses an important and realistic challenge: decision-making under partial perception loss.

+ Proposes a well-structured and interpretable control architecture (LLM-RCO) that bridges reasoning and motion planning.

+ Introduces DriveLM-Deficit, a valuable dataset for studying perception-deficit resilience.

+ Solid experimental validation across multiple agents and deficit types, with clear ablation and sensitivity analyses.

+ The “risk-averse but proactive” design philosophy is conceptually appealing and practically relevant.

**Weaknesses:**

- Limited learning novelty: The contribution lies in integration, not in representation or algorithmic innovation.

- Venue fit: The work aligns better with CoRL or ICRA, where system robustness and simulation performance are primary evaluation metrics.

- No real-world validation: Results are confined to CARLA; the framework’s performance on real sensor data or unseen environments is unknown.

- No analysis of learned representations: The paper does not examine what the fine-tuned LLM actually learns about occlusion, hazard inference, or uncertainty.

- Incomplete treatment of latency and scalability: The claimed reduction in inference frequency is not quantified.

- Safety guarantees: The LLM-generated safety constraints lack formal validation or calibration analysis.

**Questions:**

> How sensitive is the system to prompt formulations and the choice of LLM backbone (e.g., GPT-4 vs Qwen2-VL)?

> How does the Action Condition Verifier handle rapidly changing deficits (e.g., intermittent occlusions)?

> Could the authors clarify whether LLM confidence or uncertainty estimates play a role in replanning frequency or safety margins?

> What would be required to extend this method to multi-modal perception loss (camera + LiDAR)?

> Have the authors considered learning-based uncertainty modeling (e.g., conformal prediction or Bayesian calibration) as a complement to rule-based verification?

---

### Official Review · Reviewer_ESrJ · 2025-11-01

**Soundness:** 2
**Presentation:** 2
**Contribution:** 2
**Rating:** 2
**Confidence:** 4

**Summary:**

The paper proposes LLM-RCO, a modular system that uses multimodal LLMs to (1) infer hazards in occluded/deficit image regions, (2) produce short-term action–condition plans (move vs stop-observe-move), (3) verify conditions with rule checks, and (4) generate “loose” safety constraints. The authors release/construct a DriveLM-Deficit dataset (53,895 short clips) for fine-tuning Qwen2-VL and evaluate closed-loop in CARLA with Transfuser and Interfuser agents, reporting improvements on CARLA metrics under synthetic perception masking.

**Strengths:**

1. Clear problem motivation: perception deficits matter and default stops are suboptimal.
2. System modularization (hazard inference, planner, verifier, safety generator) is sensible and well described.

**Weaknesses:**

1. Dataset realism. DriveLM-Deficit is built from DriveLM-GVQA while occlusions are produced by mask/occlude operations driven by YOLO tracking, it's unclear whether the deficits represent real sensor failures (noise, glare, partial occlusion) or adversarial attacks. No transfer or real sensor experiments are provided.

2. Action mapping and control realism. The high-level action tokens map deterministically to throttle/brake parameters (Appendix Table 4). This discretization may be brittle and hides control instability introduced by wrong LLM plans (e.g., sudden deceleration to zero mapped to brake=0.8). No comparison to lower-level continuous controllers or MPC is shown.

3. Important experimental details are missing: how exactly masks are applied, how often LLM planning is called, timing of replanning, and hyperparameter settings for thresholds and LoRA ranks are scattered in appendix snippets rather than clearly summarized.

4. Potential data leakage and weak supervision. DriveLM-Deficit derives “move vs stop” labels from final-frame braking annotations and uses privileged simulator information (DriveLM-GVQA privileged labels). This trains the LLM on simulator privileged signals instead of realistic human commonsense; it risks overfitting to simulator artifacts.

5. Metric manipulation. The authors modify Infraction Score (IS) by excluding red-light and stop-sign violations in the deficit scenarios (they explicitly say they excluded those violations), which directly affects the Driving Score (DS = RC × IS) and makes comparisons unfair. Also the CARLA "game time" excludes model inference time while "system time" includes it in a second setting. This inconsistent accounting hides the practical latency cost of LLMs.

6. Heuristic thresholds and unverified safety logic. Several critical thresholds are arbitrary such as immediate-hazard ratio = 0.05, deficit spatial shift threshold. There is no formal safety verification which is unacceptable for safety claims.

**Questions:**

1. Labeling: Exactly how are move/stop labels created? Do they use privileged ground-truth object positions or only per-frame sensor images?
2. Metrics: Why were red-light/stop-sign infractions excluded from IS in deficit experiments? Provide results with the standard CARLA IS and with latency fully accounted for in all runs.
4. Statistical strength: Provide results over ≥10 randomized runs per scenario with confidence intervals and statistical tests.
5. Safety verification: How can you guarantee that LLM-generated safety constraints won’t increase risk in edge cases?

---

### Official Review · Reviewer_D8mg · 2025-11-01

**Soundness:** 3
**Presentation:** 3
**Contribution:** 3
**Rating:** 6
**Confidence:** 3

**Summary:**

This paper proposes a risk-averse framework leveraging LLM to integrate commonsense into autonomous systems with perception deficits. It includes a few key components: hazard inference, short-term NP, action condition verifier for re-planning, and safety constraints generation. The authors implemented a comprehensive experiment sets to validate the framework.

**Strengths:**

1. The paper is easy to follow, and overall well-written
2. The risk-averse framework is novel to me, and it really provides the hint to solve real-world long-tail headache problems of autonomous driving.
3. it addresses partial perception deficits by LLM — a real but under-studied AV safety problem
4. The experiments look solid and comprehensive.
5. The authors contribute to a new dataset, which could be valuable.

**Weaknesses:**

1. It is not clear to me that when this risk-averse framework will be triggered. AV perception systems have two failure modes, FN and FP, which can be told by humans, but not by the perception models. Please elaborate more on how and whether the trigger happens in what form with FN and FP.
2. It is unclear the runtime latency with LLM in the AV software loop for perception and planning.
3. The experiment is on CARLA, not on real-world vehicles.
4. Rule-Heavy Safety ComponentsAlthough pitched as LLM-based, the system still relies on: rule thresholds (e.g., 5% image area for hazard) and heuristics for action check.
5.If LLM outputs unsafe plan and verifier misses it, consequences unclear.

**Questions:**

See the weaknesses

---

### Note · Authors · 2025-11-22

I have read and agree with the venue's withdrawal policy on behalf of myself and my co-authors.